# Influence of Clinical and Surgical Factors on Uterine Carcinosarcoma Survival

**DOI:** 10.3390/cancers15051463

**Published:** 2023-02-25

**Authors:** Myriam Gracia, Yusuf Yildirim, Ronalds Macuks, Rosanna Mancari, Patriciu Achimas-Cadariu, Stephan Polterauer, Sara Iacoponi, Ignacio Zapardiel

**Affiliations:** 1Gynecologic Oncology Unit, La Paz University Hospital, Paseo de la Castellana 261, 28046 Madrid, Spain; 2Tepecik Training and Research Hospital, Izmir 35020, Turkey; 3Latvian Oncology Center, Riga Eastern Clinical University Hospital, LV-1079 Riga, Latvia; 4Division of Gynecologic Oncology, European Institute of Oncology, IRCCS, 20139 Milan, Italy; 5Gynecologic Oncology Unit, IRCCS Regina Elena National Cancer Institute, 00144 Rome, Italy; 6Prof Dr Ion Chiricuta Cluj-Napoca, University of Medicine and Pharmacy Iuliu Hatieganu, 400337 Cluj-Napoca, Romania; 7Department of Obstetrics and Gynecology, Comprehensive Cancer Center Vienna, Medical University of Vienna, 1090 Vienna, Austria

**Keywords:** uterine carcinosarcoma, prognostic factors, survival

## Abstract

**Simple Summary:**

Uterine carcinosarcoma is a rare disease that represents less than 5% of uterine malignancies, although approximately 16% of the uterine cancer-related deaths are caused by this disease. Nowadays, there seems to be a lack of prognostic stratifying studies that could lead to tailor the management of uterine carcinosarcoma. The aim of our study was to assess the impact of possible different prognostic factors on the survival of patients diagnosed with uterine carcinosarcoma. We observed that incomplete cytoreduction, presence of tumor residual after treatment, advanced FIGO stage, extrauterine disease, and tumor size were significant prognostic factors decreasing disease-free survival and overall survival.

**Abstract:**

Background: The aim of this study was to assess the impact of prognostic factors on the survival of patients diagnosed with uterine carcinosarcoma. Methods: A sub-analysis of the SARCUT study, a multicentric retrospective European study, was carried out. We selected 283 cases of diagnosed uterine carcinosarcoma for the present study. Prognosis factors influencing survival were analyzed. Results: Significant prognostic factors for overall survival were: incomplete cytoreduction (HR = 4.02; 95%CI = 2.68–6.18), FIGO stages III and IV (HR = 3.21; 95%CI = 1.83–5.61), tumor persistence after any treatment (HR = 2.90; 95%CI = 1.97–4.27), presence of extrauterine disease (HR = 2.62; 95%CI = 1.75–3.92), a positive resection margin (HR = 1.56; 95%CI = 1.05–2.34), age (HR = 1.02; 95%CI = 1.00–1.05), and tumor size (HR = 1.01; 95%CI = 1.00–1.01). Significant prognostic factors for disease-free survival were: incomplete cytoreduction (HR = 3.00; 95%CI = 1.67–5.37), tumor persistence after any treatment (HR = 2.64; 95%CI = 1.81–3.86), FIGO stages III and IV (HR = 2.33; 95%CI = 1.59–3.41), presence of extrauterine disease (HR = 2.13; 95%CI = 1.44–3.17), administration of adjuvant chemotherapy (HR = 1.84; 95%CI = 1.27–2.67), a positive resection margin (HR = 1.65; 95%CI = 1.11–2.44), presence of LVSI (HR = 1.61; 95%CI = 1.02–2.55), and tumor size (HR = 1.00; 95%CI = 1.00–1.01). Conclusions: Incomplete cytoreduction, presence of tumor residual after treatment, advanced FIGO stage, extrauterine disease, and tumor size are significant prognostic factors decreasing disease-free survival and overall survival of patients with uterine carcinosarcoma.

## 1. Introduction

Carcinosarcomas, in the past known as malignant mixed Mullerian tumors (MMMTs) and classified as uterine sarcomas, can arise in any lower female genital tract organ but are commonly originated in the uterus. Uterine carcinosarcoma (UC) is a rare disease and represents less than 5% of uterine malignancies, although approximately 16% of the uterine cancer-related deaths are cause by this disease [1]. Histologically, UC is a tumor with the presence of both carcinomatous (epithelial) and sarcomatous (stromal tissue) elements. The epithelial element, in most cases, is poorly differentiated serous carcinoma, and the sarcomatous element is used to present a high-grade sarcoma histology [2]. An increasing number of current studies support the theory that both elements originate from a carcinoma lineage that undergoes sarcomatous dedifferentiation due to the similar chromosomal abnormalities of all histological components [3,4]. Due to this, UC is considered an endometrial neoplasm, and consequently, endometrial cancer staging is applicable in terms of management, FIGO, and TNM classifications [5]. Postmenopausal age, long-term tamoxifen use, nulliparity, obesity, black race, or previous pelvic radiation are predisposing factors [5]. It shows a relatively poor prognosis compared to other epithelial endometrial malignancies. For stage I–II disease, the 5-year disease-free survival (DFS) rate is 59%, for stage III disease it is 22%, and for stage IV disease, 9% [6]. Almost 60% of patients present extrauterine disease and 10% present distant metastasis at initial diagnosis. Lung, peritoneum, and pelvic and para-aortic nodes are the most common sites [7]. 

Regarding prognostic factors, the FIGO stage at diagnosis is the most important for patients with UC. Some of the factors associated with a worse prognosis seem to be myometrial invasion, lymph vascular space invasion (LVSI), lymph node metastasis, peritoneal spread [8,9], sarcomatous component, tumor size, and distant metastasis [10,11]. 

Surgery is the cornerstone of treatment and consists of total hysterectomy, bilateral salpingo-oophorectomy, omentectomy, and resection of all gross disease. The reported lymph node metastasis rate varies between 30% and 60%; for this reason, pelvic and para-aortic lymphadenectomy are also recommended since the results not only impact staging, but may improve overall survival (OS) [1,6]. Adjuvant treatment with chemotherapy and radiotherapy could improve survival, as shown in different large studies [12,13]. 

Nowadays, there seems to be a lack of prognostic stratifying studies that could lead to tailor the management of uterine carcinosarcoma. The aim of our study was to assess the impact of possible different prognostic factors on disease-free and overall survival of patients diagnosed with uterine carcinosarcoma.

## 2. Materials and Methods

A sub-analysis of the SARCUT study (SARComa of the Uterus), which is a multicentric retrospective European study, was carried out. A total of 41 centers participated in the study. A total of 966 patients diagnosed between January 2001 and December 2007 with any type of uterine sarcoma (including carcinosarcoma) were enrolled in the study. Among them, we selected and analyzed all cases diagnosed with uterine carcinosarcoma.

The Institutional Review Board approval was obtained at the coordinating center (#PI-1382), and if requested, at the other participating centers. The study was disseminated online through European research platforms and gynecologic oncology societies. Then, a total of 53 researchers from 46 centers across Europe participated in the study.

For the present sub-analysis, the inclusion criteria included a confirmed histologic diagnosis of endometrial carcinosarcoma, FIGO stages I to IV (International Federation of Gynecology and Obstetrics 2009 staging system) [14], signed informed consent to undergo the treatment, and adequate follow-up of the patient. If any of the previous conditions were not present, the cases were excluded from the analysis.

We analyzed the variables of the study following a standardized criteria for all the participating centers. We established a uniform nomenclature for all the surgical procedures performed, sites of recurrence, and adjuvant treatment administered. The type of procedure performed depended on the extension of the disease and the surgical approach was established based on the surgeons’ preferences. Complete cytoreduction was defined by no residual tumor burden after surgery and residual disease as the persistence of tumor after any treatment [15]. Extrauterine involvement refers to abdominal disease outside the uterus. Follow-up was carried out every three months during the first year, every six months during the first five years, and annually until the end of the follow-up after ten years. At each of the visits, we performed a clinical examination, a blood sampling test, and CT-scan/MRI. To assess local or distant recurrence, a histological sample was mandatory and had to be diagnosed three months after the last treatment. Each institution administered adjuvant chemotherapy or radiotherapy based on their own protocols. Data collection was performed in a web-based encrypted database and each researcher carried out the registration of each one of their cases. 

### Statistical Analysis

Qualitative variables were presented as absolute values and proportions. Quantitative data were described as mean and standard deviation (SD). The comparison of qualitative variables between groups was carried out by the chi-square test, and between quantitative variables by Student’s t-test and the ANOVA test. Subgroup analysis was performed using Bonferroni’s post-hoc test. The multivariate analysis to determine potential risk factors was performed by means of logistic regression. Survival analyses were calculated using Kaplan–Meier curves and the Mantel–Cox method. The α-error was set at 5% and all the statistical comparisons were two-sided. Data analysis was performed with the statistical software S.P.SS 17.0 (IBM, Armonk, NY, USA). 

## 3. Results

A total of 283 patients with uterine carcinosarcoma who met the inclusion criteria were identified. The patients’ baseline characteristics and treatment modalities are summarized in Table 1. 

The mean (±SD) age of the patients was 66.5 ± 10.7 years, with 243 (85.9%) postmenopausal women. Most of the patients presented symptoms at diagnosis, with 222 (78.4%) women presenting vaginal bleeding. A total of 171 (60.4%) patients were diagnosed at stage I–II of the disease, and 261 (92.2%) patients underwent open surgery, where hysterectomy with or without bilateral salpingo-oophorectomy was the most frequent surgical procedure performed. Complete cytoreduction was achieved in 177 (72.5%) patients. Among all patients, 171 (60.5%) women received radiotherapy and 103 (36.4%) received chemotherapy. 

The mean (±SD) follow-up time of the cohort was 42 ± 37 months. The mean (±SD) overall survival time was 36 ± 35.1 months, and 102 (36%) patients died during follow-up as a result of the disease. The mean (±SD) time to relapse was 33 ± 16 months. During the follow-up time, 112 (39.6%) patients experienced a recurrence; among them, 73 (25.8%) patients presented pelvic relapse and 88 (31.1%) suffered distant metastatic recurrence. OS at 5 and 10 years was 56.5% and 40.8%, respectively (Figure 1). The 5- and 10-year DFS was 50.2% and 46.1%, respectively (Figure 2).

To identify all prognosis factors associated with OS and DFS, a multivariate analysis was conducted. All independent significant factors related to OS, DFS, pelvic relapse, and metastatic relapse are shown in Table 2. 

The factors negatively impacting OS were: incomplete cytoreduction (HR = 4.02; 95%CI = 2.68–6.18), FIGO stage III and IV (HR = 3.21; 95%CI = 1.83–5.61), tumor persistence after any treatment (HR = 2.90; 95%CI = 1.97–4.27), presence of extrauterine disease (HR = 2.62; 95%CI = 1.75–3.92), positive resection margins (HR = 1.56; 95%CI = 1.05–2.34), age (HR = 1.02; 95%CI = 1.00–1.05), and tumor size (HR = 1.01; 95%CI = 1.00–1.01). 

Significant prognostic factors for disease-free survival were: incomplete cytoreduction (HR = 3.00; 95%CI = 1.67–5.37), tumor persistence after any treatment (HR = 2.64; 95%CI = 1.81–3.86), FIGO stage III and IV (HR = 2.33; 95%CI = 1.59–3.41), presence of extrauterine disease (HR = 2.13; 95%CI = 1.44–3.17), administration of adjuvant chemotherapy (HR = 1.84; 95%CI = 1.27–2.67), positive resection margins (HR = 1.65; 95%CI = 1.11–2.44), presence of LVSI (HR = 1.61; 95%CI = 1.02–2.55), and tumor size (HR = 1.00; 95%CI = 1.00–1.01). 

The most important factors related to pelvic recurrence were: incomplete cytoreduction (HR = 2.75; 95%CI = 1.49–5.10), tumor persistence after any treatment (HR: 2.51; 95%CI = 1.54–4.09), administration of adjuvant chemotherapy (HR = 2.07; 95%CI = 1.29–3.31), and positive resection margins (HR = 2.04; 95%CI = 1.27–3.28). We also found that presence of lymph vascular invasion and extrauterine involvement were factors that significantly increased the risk of pelvic recurrence (*p* < 0.05). 

The most important factor significantly associated with metastatic relapse was incomplete cytoreduction (HR = 5.41; 95%CI = 2.73–10.70). Advanced FIGO stage (HR = 2.47; 95%CI = 1.35–4.50) and positive resection margins (HR = 2.00; 95%CI = 1.25–3.19) were also important factors associated with distant relapse. Other factors, such as presence of extrauterine disease and tumor persistence after any treatment or tumor size, were associated with poor prognosis due to increased metastatic relapses (*p* < 0.05). 

We did not observe a significant association between adjuvant radiotherapy administration, presence of necrosis, surgical procedure, or route of approach with DFS or OS in the multivariate analysis. 

## 4. Discussion

The rarity of UC means that experience with this tumor type is usually low and most of the studies have been single institutional studies involving a small sample of patients, and therefore definite information regarding prognostic factors remains unclear. Despite progression in molecular classification over the past years in endometrial carcinoma, the best protocol treatment and prognosis factors related to UC are still lacking [16]. This is one of the main reasons to carry out the present study, where the most important clinical and surgical factors that influence DFS and OS among women with uterine carcinosarcoma have been presented, and which reports one of the largest series in the literature. 

UC is a relatively rare tumor that has a more aggressive behavior compared to other endometrial carcinomas. Including all FIGO stages, we observed a 5- and 10-year DFS of 50.2% and 46.1%, respectively. The OS at 5 and 10 years was 56.5% and 40.8%, respectively. In terms of survival, our results showed a better prognosis compared to other series that reported a 5-year OS and DFS of 30% and 27%, respectively [17]. 

Our results revealed that incomplete cytoreduction is the most important prognostic factor related to DFS and OS (*p* < 0.001). In the same line, other factors related to tumor burden, such as residual disease after any kind of treatment or positive resection margins after surgery, are important factors that significantly decrease DFS, affecting both pelvic and distant relapses, and OS. In patients with advanced UC, gross total resection of disease should be the goal of primary cytoreductive surgery. A large multicenter study of 486 women with stages I–IV UC found that a residual tumor size > 1 cm was associated with a shorter OS [18]. Similarly, previous studies have also observed an improvement in OS when no residual tumor is achieved in stages III–IV of UC [19,20,21]. Despite these results, Harano et al. suggested that the benefit of cytoreduction in patients with UC is not well-known and the cutoff for residual disease has not still been established. The reason for this hypothesis is that their study, as ours, included early-stage disease, and this population could have modified the impact of the surgery on the oncological outcomes [18]. On the contrary, other authors found that the extent of surgery did not have a statistically significant effect on DFS or OS after multivariate analysis [17,22]. One of the reasons for this result might be the small sample size and low number of patients with complete cytoreductive surgery. 

We found that 39.6% of our patients presented an advanced stage of the disease (FIGO stages III–IV) and that 28.3% had extrauterine involvement of the disease, a proportion slightly lower compared to other studies, which found more than 60% of patients with stage III or IV [22]. In our cohort, these two factors were associated with a decrease of OS and DFS, including pelvic and metastatic relapses. In fact, there is a general agreement that the surgical stage in UC could be the most important prognostic indicator regardless of how the patient was staged [23]. In addition, advanced stages have also been associated with poor prognosis in other retrospective studies [24].

We also found that tumor size was associated with a worse prognosis for both OS and DFS, specifically increasing the risk of distant metastasis but not pelvic recurrence. This is probably because in large tumors, the risk of extrauterine disease and incomplete cytoreduction is high. However, this result is not consistent in all studies since there are some authors that did not find a relationship between tumor size and survival in patients with carcinosarcoma [25].

The inverse relationship between LVSI and survival has been reported in previous studies. In our cohort, LVSI was associated with a shorter DFS and with an increased rate of pelvic recurrence, but there was no statistically significant association between LVSI and OS or distant metastasis. This result is consistent with a study that found a significant association between LVSI and DFS, but without any impact on OS; however, in the advanced stage, LVSI did not play any role in the oncological outcomes, since other factors could have a more predominant importance [18,21]. Finally, there are other authors that did not find any significant relation between LVSI and survival in UC [22,25]. 

Several studies included age as a prognostic factor in the survival of patients with UC, but results are inconsistent and heterogeneous. Our data showed that age is an independent prognostic factor for OS but has no implication in DFS. The same results are reported by some studies [25,26], but contrasting results for others [18,22]. Due to this, from our point of view, the role of age seems to remain unclear.

Although the management of uterine carcinosarcoma is mainly surgical, recent investigations propose a multimodality treatment approach, including adjuvant chemotherapy or radiotherapy after surgery [26]. A total of 103 (36.4%) and 171 (60.4%) of our patients received adjuvant chemotherapy or radiotherapy, respectively. Among them, there was a high proportion of patients receiving chemotherapy with an advanced stage of the disease (62% stages III–IV), but in contrast, the majority (67%) of patients receiving adjuvant radiotherapy presented an early-stage disease. In our study, the administration of chemotherapy did not affect OS or distant relapses but negatively impacted DFS and pelvic recurrences. We are not able to explain this because chemotherapy was administered mainly in advanced-stage patients, which is associated with a worse prognosis, which would also decrease OS, not only DFS. Adjuvant radiotherapy did not impact survival rates, as in other recent studies [26]. 

Therapeutic strategies in the adjuvant setting remain controversial for UC. To date, there are no prospective trials stating that adjuvant treatment with chemotherapy or radiotherapy confers an overall survival benefit in patients with uterine carcinosarcoma. There is only one phase 3 trial fully dedicated to UC, and it did not find a statistically significant advantage in DFS or OS for adjuvant chemotherapy over radiotherapy [27]. 

Several retrospective studies suggest that adjuvant multimodal treatment, consisting of sequential chemotherapy and radiotherapy, could improve OS in both early and advanced stages compared to observation [28,29]. However, no prospective randomized trials have been performed to validate these observations. 

The main strength of our study is the large sample of patients with this rare disease and the multicenter nature of the study. There are also several limitations and weaknesses. First, it was a retrospective study, which could lead to some biases; second, some parameters, such as the histologic subtype of the epithelial and sarcomatous elements, and the molecular characteristics were not recorded; finally, there could have been differences in management modalities over the period of the study and among centers, which could influence the results, although at the same time, this could lead to a good external validity of the study. 

## 5. Conclusions

Our study showed that incomplete cytoreduction, presence of tumor residual after treatment, advanced FIGO stage, extrauterine disease, and tumor size are significant prognostic factors that impact disease-free survival and overall survival of patients with uterine carcinosarcoma. In addition, advanced age exclusively decreased overall survival, and the presence of lymph vascular space invasion and adjuvant chemotherapy were related to a decrease in disease-free survival. 

## Figures and Tables

**Figure 1 cancers-15-01463-f001:**
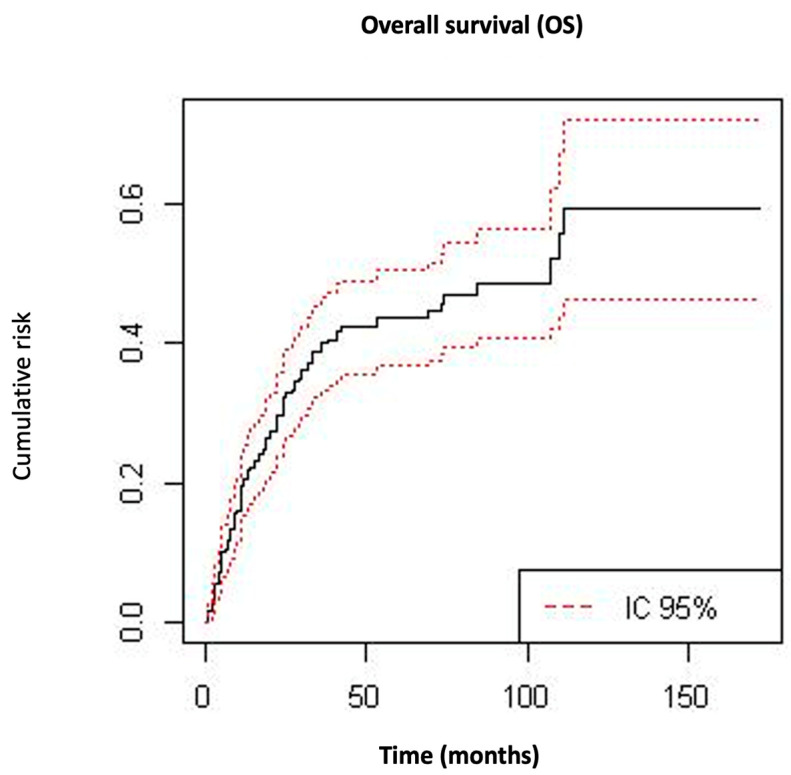
Kaplan–Meier curves for overall survival.

**Figure 2 cancers-15-01463-f002:**
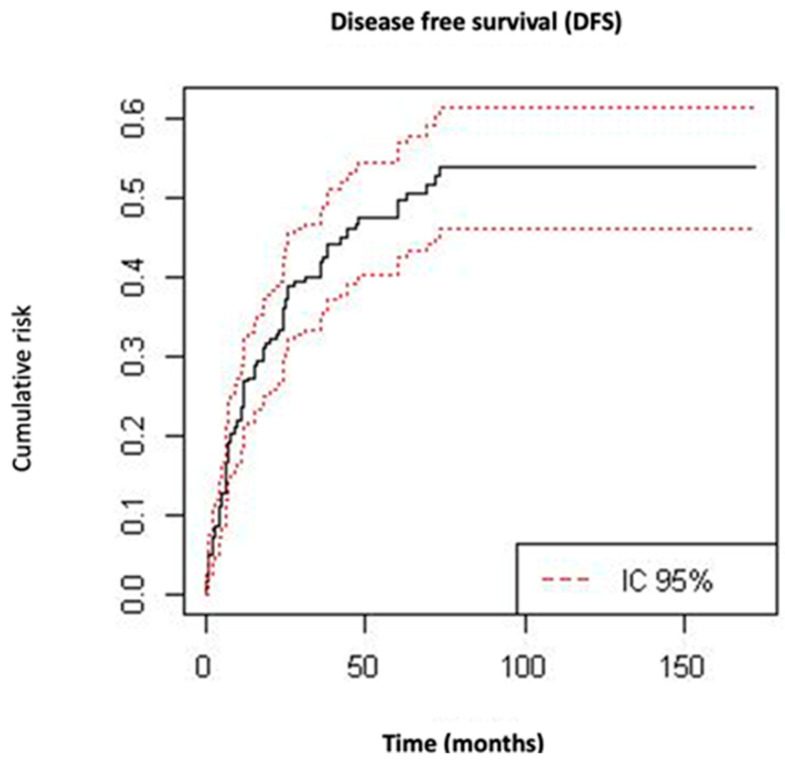
Kaplan–Meier curves for disease-free survival.

**Table 1 cancers-15-01463-t001:** Baseline characteristics and treatment modalities of patients with carcinosarcoma.

Baseline Characteristics	Number of Cases (n = 283)
Age, years (mean ± SD)	66.55 ± 10.7
Menopause	243 (85.9%)
Smoker	16 (5.7%)
Parity, births (mean ± SD)	2.7 ± 2.1
Previous pelvic radiation	12 (4.2%)
Previous use of tamoxifen	13 (4.6%)
Symptomatology	
Pelvic mass	19 (6.7%)
Bleeding	222 (78.4%)
None	42 (14.8%)
FIGO Stage	
I	146 (51.6%)
II	25 (8.8%)
III	75 (26.5%)
IV	37 (13.1%)
Lymph vascular invasion	66 (23.3%)
Positive pelvic lymph nodes	23 (8.1%)
Positive aortic lymph nodes	8 (3%)
Positive resection margins	84 (29.7%)
Tumor size, mm (mean ± SD)	56 ± 39.5
Necrosis	77 (27.2%)
Estrogen receptors	11 (3.9%)
Extrauterine involvement	80 (28.3%)
Surgical approach	
Laparoscopy	9 (3.2%)
Laparotomy	261 (92.2%)
Vaginal	6 (2.1%)
Surgical procedure	
Hysterectomy	270 (95.4%)
Bilateral salpingo-oophorectomy	261 (92.2%)
Omentectomy	76 (26.9%)
Pelvic lymphadenectomy	109 (38.5%)
Para-aortic lymphadenectomy	61 (21.6%)
Appendectomy	11 (3.9%)
Recto-sigmoid resection	3 (1.1%)
Ureteral resection	1 (0.4%)
Vascular resection	1 (0.4%)
Small bowel resection	3 (1.1%)
Surgical cytoreduction	
Complete	177 (62.5%)
Minimal residual disease (≤1 cm)	19 (6.7%)
Gross residual disease (>1 cm)	32 (11.3%)
Unknown	55 (19.4%)
Radiotherapy	171 (60.4%)
Chemotherapy	103 (36.4%)

**Table 2 cancers-15-01463-t002:** Multivariate analysis of independent prognostic factors related to overall survival, disease-free survival, pelvic relapse, and metastatic relapse in patients with carcinosarcoma (HR: hazard ratio; CI: confidence interval; LVSI: lymph vascular space invasion; residual disease: tumor persistence after any treatment).

	Prognostic Factor	HR (95% CI)	*p*
Overall survival	Incomplete cytoreduction	4.02 (2.62–6.18)	<0.001
	FIGO stage III–IV	3.21 (1.83–5.61)	<0.001
	Residual disease	2.90 (1.97–4.27)	<0.001
	Extrauterine disease	2.62 (1.75–3.92)	<0.001
	Positive resection margins	1.56 (1.05–2.34)	0.002
	Age	1.02 (1.00–1.05)	0.009
	Tumor size	1.01 (1.00–1.01)	<0.001
Disease-free survival	Incomplete cytoreduction	3.00 (1.67–5.37)	<0.001
	Residual disease	2.64 (1.81–3.86)	<0.001
	FIGO stage III–IV	2.33 (1.59–3.41)	<0.001
	Extrauterine disease	2.13 (1.44–3.17)	<0.001
	Adjuvant chemotherapy	1.84 (1.27–2.67)	0.001
	Positive resection margins	1.65 (1.11–2.44)	0.012
	LVSI	1.61 (1.02–2.55)	0.039
	Tumor size	1.00 (1.00–1.01)	<0.001
Pelvic recurrence	Incomplete cytoreduction	2.75 (1.49–5.10)	0.001
	Residual disease	2.51 (1.54–4.09)	<0.001
	Adjuvant chemotherapy	2.07 (1.29–3.31)	0.002
	Positive resection margins	2.04 (1.27–3.28)	0.003
	LVSI	1.95 (1.08–3.51)	0.024
	Extrauterine disease	1.89 (1.14–3.14)	0.013
Metastasis	Incomplete cytoreduction	5.41 (2.73–10.70)	<0.001
	FIGO stage III–IV	2.47 (1.35–4.50)	0.003
	Positive resection margins	2.00 (1.25–3.19)	0.003
	Extrauterine disease	1.97 (1.20–3.22)	0.006
	Residual disease	1.78 (1.08–2.92)	0.022
	Tumor size	1.00 (1.00–1.01)	0.003

## Data Availability

Data for quality control could be available upon request to the corresponding author.

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
