# Peer review of "Influence of Clinical and Surgical Factors on Uterine Carcinosarcoma Survival"

_cancers, 2023, doi:10.3390/cancers15051463_

Round 1

Reviewer 1 Report

Very interesting paper accumulated 283 cases and a large number of carcinosarcoma cases. The authors report that the outcome of complete surgical cytoreduction is a prognostic factor. How many institutions did the 283 patients gather? How many cases per institution? Were there any differences in treatment between institutions? For example, what was the difference in outcome between institutions that treated more than 10 cases and those that treated less than 10 cases?

Author Response

Thank you for your comments. 

  • First comment: a total of 41 centers participated in the study. We han added it in the manuscript.
  • Second comment: 6.9 cases per center. 
  • Third comment: surgical treatment dependent on the extension of the disease and adjuvant treatment was based on each institution´s protocol. It is mentioned in material and methods section (Line 107 and 116).  
  • Fourth comment: Unfortunately we did not analyzed outcomes between institutions. From our point of view, we considered that complete cytoreduction rate is a more direct marker than the number of cases per center. 

Reviewer 2 Report

retrospective analysis of a series of carcinosarcoma

+ large series of patients 

- no pathological review, no pathological details (heterologous component by example), no molecular classification

the analysis finds classical prognostic factors. however, this large multicenter retrospective analysis should be published with just adding some sentences about the pathological & molecular limitations of this work 

Author Response

Thank you for your comments, we agree with you about molecular and pathological limitations of the study, but we have made reference to this consideration in line 267-270 in the main manuscript.